# Individual Lymphocyte Sensitivity to Steroids as a Reliable Biomarker for Clinical Outcome after Steroid Withdrawal in Japanese Renal Transplantation

**DOI:** 10.3390/jcm10081670

**Published:** 2021-04-13

**Authors:** Masaaki Okihara, Hironori Takeuchi, Yukiko Kikuchi, Isao Akashi, Yu Kihara, Osamu Konno, Hitoshi Iwamoto, Takashi Oda, Sachiko Tanaka, Sakae Unezaki, Toshihiko Hirano

**Affiliations:** 1Department of Kidney Transplantation Surgery, Tokyo Medical University Hachioji Medical Center, 1163 Tatemachi, Hachioji-shi, Tokyo 193-0998, Japan; mokihara@tokyo-med.ac.jp (M.O.); isaoakashi@gmail.com (I.A.); yukihara7@gmail.com (Y.K.); konno@tokyo-med.ac.jp (O.K.); hitoiwa@tokyo-med.ac.jp (H.I.); 2Department of Pharmacy, Tokyo Medical University Hospital, 6-7-1 Nishishinjuku, Shinjuku-ku, Tokyo 160-0023, Japan; h-take@tokyo-med.ac.jp; 3Department of Practical Pharmacy, Tokyo University of Pharmacy and Life Sciences, 1432-1 Horinouchi, Hachioji, Tokyo 192-0392, Japan; kiku2kyoto801@yahoo.co.jp (Y.K.); unezaki@toyaku.ac.jp (S.U.); 4Department of Nephrology, Tokyo Medical University Hachioji Medical Center, 1163 Tatemachi, Hachioji-shi, Tokyo 193-0998, Japan; takashio@tokyo-med.ac.jp; 5Clinical Pharmacology, Tokyo University of Pharmacy and Life Sciences, 1432-1 Horinouchi, Hachioji, Tokyo 192-0392, Japan; sachiko@toyaku.ac.jp

**Keywords:** steroid withdrawal, steroid reduction, lymphocyte sensitivity, biomarker, renal transplantation

## Abstract

Recently, steroid reduction/withdrawal regimens have been attempted to minimize the side effects of steroids in renal transplantation. However, some recipients have experienced an increase/resumption of steroid administrations and acute graft rejection (AR). Therefore, we investigated the relationship between the individual lymphocyte sensitivity to steroids and the clinical outcome after steroid reduction/withdrawal. We cultured peripheral blood mononuclear cells (PBMCs) isolated from 24 recipients with concanavalin A (Con A) in the presence of methylprednisolone (MPSL) or cortisol (COR) for four days, and the 50% of PBMC proliferation (IC_50_) values and the PBMC sensitivity to steroids were calculated. Regarding the experience of steroid increase/resumption and incidence of AR within one year of steroid reduction/withdrawal, the IC_50_ values of these drugs before transplantation in the clinical event group were significantly higher than those in the event-free group. The cumulative incidence of steroid increase/resumption and AR in the PBMC high-sensitivity groups to these drugs before transplantation were significantly lower than those in the low-sensitivity groups. These observations suggested that an individual’s lymphocyte sensitivity to steroids could be a reliable biomarker to predict the clinical outcome after steroid reduction/withdrawal and to select the patients whose dose of steroids can be decreased and/or withdrawn after transplantation.

## 1. Introduction

Immunosuppressive drugs and the therapies based on these drugs have markedly improved the graft survival and function, which has enabled successful renal transplantation. Currently, calcineurin inhibitors (CNIs), steroids (glucocorticoids: GCs), mycophenolate mofetil (MMF), basiliximab (Bx), and everolimus (EVL) are used in combination at renal transplant centers to reduce the dose of each drug and side effects while maintaining immunosuppressive effects. GCs have been used since the earliest renal transplantations, while the long-term use of GCs is known to be associated with serious complications including hypertension, hyperlipidemia, glucose intolerance, cataracts, and loss of bone mineral density [1]. Therefore, GC reduction/withdrawal after renal transplantation has been attempted in recent times. Several studies have shown that GC reduction/withdrawal could be a safe standard for immunosuppressive therapy in low immunological risk recipients [2,3,4,5,6,7]. On the other hand, some studies have shown that GC reduction/withdrawal was associated with an increase in the incidence of acute graft rejection (AR) and a more rapid deterioration of graft function [8,9,10]. Thus, the problem is that there is no clear index to predict patients who can have their dose of GCs safely reduced and/or withdrawn at present.

It is well known that there could be a strong relationship between blood concentration and the therapeutic efficacy of GCs, while large individual differences in pharmacodynamics of GCs are often observed [11]. In our previous study, we focused on individual lymphocyte sensitivity to GCs as an index to predict the pharmacodynamic efficacy of GCs in renal transplantation [11]. We have also suggested that individual lymphocyte sensitivity to GCs would be a useful biomarker for GC reduction/withdrawal, because we demonstrated that in long-term stable renal transplantation, the recipients exhibiting high-sensitivity to cortisol (COR) showed significantly lower rates of AR and GC increase/resumption in serum creatinine (S–Cr) levels [12].

In the present study, we investigated the relationship between the lymphocyte sensitivity to GCs and clinical outcomes after GC reduction/withdrawal using peripheral blood mononuclear cells (PBMCs) obtained from renal transplant recipients. Based on the observation of these examinations, we discussed the usefulness of PBMC sensitivity to GCs as a reliable biomarker for the safe reduction/withdrawal of GCs within two months after renal transplantation.

## 2. Materials and Methods

### 2.1. Reagents

A Roswell Park Memorial Institute (RPMI)-1640 culture medium and fetal bovine serum were purchased from Gibco BRL (Grand Island, NY, USA). Con A was purchased from Seikaguku Kogyo Co., Tokyo, Japan. EVL and tacrolimus (TAC) were purchased from Fujifilm Wako Pure Chemical Co. (Osaka, Japan) and dissolved in ethanol. The working concentrations were prepared after dilution with ethanol. All other regents were of the best available grade.

### 2.2. Subjects

Among 29 recipients who received renal transplantation at our center from August 2008 to March 2012 and were evaluated for their PBMC sensitivity to GCs, 24 recipients (17 males and 7 females) who had their dose of methylprednisolone (MPSL) reduced to 4 mg/day within 2 months after renal transplantation were included. The recipient characteristics are shown in Table 1.

We monitored the peripheral blood mononuclear cell (PBMC) sensitivity to two glucocorticoids (GCs)—the endogenous GC cortisol (COR) and the synthetic GC MPSL.

### 2.3. GC Withdrawal Protocol

Recipients were orally administered MPSL from 2 days before transplantation, followed by intravenous injection (i.v.) administrations at dose of 1000 mg perioperatively, 250 mg postoperatively, 125 mg at day 1 (D1), and 80 mg at D2. Then, the recipients were orally administered MPSL at a dose of 60 mg/day, and the dose was tapered gradually until it reached 2 mg/day at D51. MPSL administration was stopped by the end of 3 months after transplantation if the following clinical conditions were obtained: (i) the original renal disease was unlikely to cause post-transplant recurrent glomerulonephritis, (ii) AR was not proven by the protocol renal biopsy at 3 months after transplantation, and (iii) the recipients had no experiences of AR within 3 months after transplantation.

### 2.4. PBMC Isolation and Culture

The venous bloods were collected 4 days before transplantation and 2 months after transplantation before the immunosuppressive-drug administration. Ten milliliters of blood were loaded onto 4 mL Ficoll-Hypaque lymphocyte separation solutions (Nakarai Co., Tokyo, Japan) and centrifuged at 1300× *g* for 20 min. The PBMC layer was transferred to another tube, and 5 mL of the RPMI-1640 medium supplemented with 10% fetal bovine serum, 100,000 IU/L penicillin, and 100 mg/L streptomycin were added and mixed well. Then the cells were centrifuged at 1300× *g* for 20 min. After removing the supernatant, the RPMI-1640 medium was added, and the cell suspension was subsequently mixed and centrifuged. Finally, PBMCs were diluted to 1 × 106 cells/mL with the medium. Then, 195 µL of the PBMC suspension and 1 µL of 1 mg/mL Con A solution were added to each well of the 96-well-plates, and 4 µL of ethanol as a control or the same volume of ethanol solution of GCs was added to give a total volume of 200 µL. Thus, the final concentrations of COR and MPSL were 1–100,000 and 0.1–10,000 ng/mL, respectively. After mixing, the plate was cultured at 37 °C with 5% CO_2_ for 72 h. Subsequently, a 0.5 µL of 3H-thymidine solution (18.5 KBq/well) was added into each well, and the plate was further cultured for 20 h.

### 2.5. Evaluation of the Effect of GCs on PBMC Proliferation Rate

After culturing, the cells were harvested, and the radioactivity of 3H-thymidine incorporated into PBMCs was measured using a liquid scintillation counter. The proliferation rate of PBMCs stimulated by Con A was calculated from the following formula:$$\mathrm{The}\;\mathrm{proliferation}\;\mathrm{rate}\;\mathrm{of}\;\mathrm{PBMCs}\;(\%)\;=\;\left(\frac{E_{2}-E_{0}}{E_{1}-E_{0}}\right)\;\times \;100$$
where *E*_0_, *E*_1_ and *E*_2_ represent the radioactivity incorporated into unstimulated PBMCs without drug (dpm), the radioactivity incorporated into PBMCs stimulated by Con A in the absence of drug (dpm), and the radioactivity incorporated into PBMCs stimulated by Con A in the presence of drug (dpm), respectively.

The concentration of agent that could inhibit 50% of PBMC proliferation (IC_50_) was determined from the concentration–response curve with the PBMC proliferation plotted on the vertical axis and the drug concentration plotted on the horizontal axis. These curves were made using the Emax model.

PBMC sensitivity to GCs was evaluated before the immunosuppressive drug administrations and 2 months after renal transplantation.

### 2.6. Comparison of the IC_50_ Values between the “Clinical Event Group” and the “Event-Free Group”

We compared the IC_50_ values of COR and MPSL before and after transplantation between the “clinical event group” and the “event-free group” regarding the experience of the increase/resumption of GC administrations and the incidence of AR within 1 year of GC reduction/withdrawal. We show the representative plots as supplemental data, as distributed in Figure 1.

### 2.7. Comparison of the Clinical Outcome between the PBMC High-Sensitivity Group and the PBMC Low-Sensitivity Group to GCs

A receiver operating characteristic (ROC) curve was constructed to evaluate the optimal cut-off value of dividing into the PBMC high-sensitivity group and the PBMC low-sensitivity group to GCs using our previous results regarding S–Cr levels after GC reduction/withdrawal in long-term stable renal transplant recipients [12], which found that the cut-off values of COR and MPSL were 3579.98 and 21.5 ng/mL, respectively. Regarding PBMC sensitivities to these drugs before and after transplantation, recipients in whom the IC_50_ values were less than the cut-off values were classified into the high-sensitivity group, and those in whom the IC_50_ values were more than the cut-off values were classified into the low-sensitivity group.

We compared the clinical outcomes estimated by the cumulative incidence and the onset time of increase/resumption of GC administrations and AR within 1 year of GC reduction/withdrawal between the two recipient groups. In addition, as an evaluation of renal function, we compared the changes in S–Cr levels for half a year after GC reduction/withdrawal between the two recipient groups.

### 2.8. Statistical Analysis

Statistical analyses were conducted using JMP^®^ 11 (SAS Institute Inc., Cary, NC, USA). A Wilcoxon signed-rank test was used to analyze differences in the IC_50_ values between the clinical event group and the event-free group and differences in change of S–Cr levels between the PBMC high-sensitivity group and the PBMC low-sensitivity group. Chi-squared analysis was used to examine differences in the incidence of events between any two patient subgroups. The Kaplan–Meier method was used to represent cumulative incidence curves, and the log-rank test was used to compare the cumulative incidences between two recipient groups. Fisher’s exact probability test was used to compare the onset times of clinical event between the two recipient groups. Differences were considered to be statistically significant for values of *p* < 0.05.

## 3. Results

### 3.1. The Recipient Characteristics

We compared the recipient characteristics between the PBMC high-sensitivity group and the PBMC low-sensitivity group. The differences between two recipient groups were not statistically significant, including differences of the immunosuppressive drugs other than MPSL, in all characteristics.

### 3.2. Comparison of the IC_50_ Values between the Clinical Event Group and the Event-Free Group Regarding Experience of Increase/Resumption of GC Administrations

We compared the IC_50_ values of COR and MPSL on the mitogen-activated proliferation of PBMCs before and after transplantation between the clinical event group and the event-free group regarding an experience of increase in dose and/or resumption of GC administration within one year of GC reduction/withdrawal, as shown in Table 2. The recipients who experienced a GC increase/resumption had significantly higher IC_50_ values of these drugs before transplantation than those in recipients without events (*p* = 0.042 and 0.0049, respectively).

We divided the recipients into two subgroups, i.e., the PBMC high-sensitivity patients and the PBMC low-sensitivity patients according to the median IC_50_ values of GCs on the mitogen-activated proliferation of PBMCs, as described above (see Materials and Methods). Then, we compared the incidences of increase/resumption of GC administrations between the two subgroups. The recipients exhibiting higher IC_50_ values to these drugs before and after transplantation showed significantly higher incidence of increase/resumption of GC administrations (*p* < 0.05). Thus, the GC high-sensitivity recipients, regardless of blood sampling time, showed significantly lower incidence of increase/resumption of GC administrations.

### 3.3. Comparison of the IC_50_ Values between the Clinical Event Group and the Event-Free Group Regarding Incidence of AR

We compared the IC_50_ values of COR and MPSL before and after transplantation between the clinical event group and the event-free group regarding incidence of AR within one year of GC reduction/withdrawal, as shown in Table 3. The recipients who experienced AR showed significantly higher IC_50_ values of these drugs before transplantation than those in recipients without AR (*p* = 0.011 and 0.0071, respectively). Though the difference was not significant, the IC_50_ values of these drugs after transplantation in the event group tended to be higher than those in the event-free group (*p* = 0.083 and 0.08, respectively).

Similarly to the observations described in Section 1, we compared incidences of AR between the two recipient subgroups divided according to the PBMC sensitivity to GCs. The recipients exhibiting higher IC_50_ values to these drugs before transplantation showed a significantly higher incidence of AR (*p* < 0.05). Thus, the high GC sensitivity recipients before transplantation showed a significantly lower incidence of AR. In addition, the incidences of AR in the recipients a exhibiting high GC sensitivity were significantly lower, regardless of blood sampling time, than those in the recipients exhibiting a low GC sensitivity (*p* < 0.05).

### 3.4. Comparison of the Cumulative Incidence of GC Increase/Resumption between the PBMC High-Sensitivity Group and the PBMC Low-Sensitivity Group to GCs

We compared the cumulative incidences of the increase/resumption of GC administrations within one year of GC reduction/withdrawal between the PBMC high-sensitivity group and the PBMC low-sensitivity group to COR and MPSL before and after transplantation, as shown in Figure 2. The recipients exhibiting a high sensitivity to these drugs before transplantation and the recipients exhibiting a high sensitivity to COR after transplantation could maintain their allograft function on GC reduction/withdrawal (Figure 2A–C). The recipients exhibiting a low sensitivity to GC experienced higher incidences of GC increase/resumption within one year of GC reduction/withdrawal than the high-sensitivity recipients in all cases.

### 3.5. Comparison of the Cumulative Incidence of AR between the PBMC High-Sensitivity Group and the PBMC Low-Sensitivity Group to GCs

We compared the cumulative incidences of AR within one year of GC reduction/withdrawal between the PBMC high-sensitivity group and the PBMC low-sensitivity group to COR and MPSL before and after transplantation, as shown in Figure 3. The recipients exhibiting a low sensitivity to these drugs before transplantation showed a significantly higher incidence of AR within one year of GC reduction/withdrawal compared to that in the recipients exhibiting a high GC sensitivity (*p* = 0.0028 and 0.003, respectively) (Figure 3A,B). Thus, the results suggested that the low-sensitivity group experienced a higher incidence of AR within one year of GC reduction/withdrawal than the high-sensitivity group.

### 3.6. Comparison of the Onset Time of GC Increase/Resumption between the PBMC High-Sensitivity Group and the PBMC Low-Sensitivity Group to GCs

We compared the onset times of the increase/resumption of GC administrations after GC reduction/withdrawal between the PBMC high-sensitivity group and the PBMC low-sensitivity group to COR and MPSL before and after transplantation, as shown in Figure 4. The recipients exhibiting a low COR sensitivity after transplantation showed a significantly higher incidence of GC increase/resumption within one month of GC reduction/withdrawal compared to that in the recipients exhibiting a high GC sensitivity (*p* = 0.032) (Figure 4C). In addition, the most frequent onset times of GC increase/resumption were within one month of GC reduction/withdrawal in the low-sensitivity group and one-to-two months after GC reduction/withdrawal in the high-sensitivity group, although the difference between two recipient groups was not statistically significant.

### 3.7. Comparison of the Onset Time of AR between the PBMC High-Sensitivity Group and the PBMC Low-Sensitivity Group to GCs

We compared the onset times of AR after GC reduction/withdrawal between the PBMC high-sensitivity group and the PBMC low-sensitivity group to COR and MPSL before and after transplantation, as shown in Figure 5. The most frequent onset times of AR were similar to the results shown above in the two recipient groups (see Section 5), and no significant difference was observed between the two recipient groups.

### 3.8. Comparison of the Change in S–Cr Levels between the PBMC High-Sensitivity Group and the PBMC Low-Sensitivity Group to GCs

We compared the changes in S–Cr levels for half a year after GC reduction/withdrawal between the PBMC high-sensitivity group and the PBMC low-sensitivity group to COR and MPSL before and after transplantation, as shown in Figure 6. No significant differences were observed between any two recipient groups for half a year.

## 4. Discussion

GC maintenance therapy is still extensively used in the majority of renal transplant centers across the world. Recently, however, a shift toward GC decrease/withdrawal to avoid side effects has been gathering momentum. While some recipients can have their dose of GCs reduced or withdrawn without major problems, others need to have their dose of GC increased or resumed to maintain renal function and prevent AR [2,3,4,5,6,7,8,9,10]. Therefore, we wondered whether GC sensitivity affected the clinical outcome and then investigated the relationship between GC sensitivity and clinical outcome after GC reduction/withdrawal.

In the present study, the IC_50_ values of GCs in the clinical event group were higher than those in the event-free group, suggesting that GC sensitivity and the occurrence of clinical events showed significant associations. In addition, the cumulative incidences of events in the GC high-sensitivity groups before transplantation were significantly lower than those in the GC low-sensitivity groups. These observations suggested that individual GC sensitivity before transplantation could be a reliable biomarker for safe GC decrease/withdrawal after transplantation.

In the present study, although differences in the onset time of events between the two recipient groups were not significant, the GC low-sensitivity recipients showed a high incidence of the events within one month of GC reduction/withdrawal. The immunosuppressive effect of Bx was reported to continue for 51 ± 9 days in adult recipients [13]. At one month after GC reduction/withdrawal (which corresponds to two-to-three months after the transplantation), the attenuation and disappearance of the immunosuppressive efficacy of Bx, the insufficient immunosuppressive effect of GCs due to GC reduction/withdrawal, and a GC low-sensitivity were considered to be the causes of AR incidence. The GC low-sensitivity groups in long-term stable renal transplant recipients have been reported to show relatively high incidences of clinical events at four months after GC reduction/withdrawal [12]. These findings, together with our present data, suggested that we need to be careful for GC reduction/withdrawal because the most frequent onset time of AR was different depending on the periods of GC reduction/withdrawal.

The reasons why recipients exhibiting a high sensitivity to GCs had good clinical outcomes after GC reduction/withdrawal were that it will be easy to reduce the dose of MPSL for recipients with a high MPSL sensitivity and immune cell activation by an allograft in recipients with a high COR sensitivity could be suppressed enough by endogenous COR after MPSL reduction/withdrawal.

In the present study, we observed that GC sensitivity before transplantation was higher than that after transplantation. The combination of GCs with other immunosuppressive drugs might affect GC sensitivity, which made it difficult to evaluate GC sensitivity. Therefore, it is recommended to estimate GC sensitivity before immunosuppressive drug administration.

Though original renal diseases including highly recurrent glomerulonephritis such as immunoglobulin A (IgA) nephropathy were not taken into account for the analysis of the present study, other studies have reported that the recipients who had their dose of GCs decreased and/or withdrawn in the early stages of renal transplantation showed higher rates of recurrence and graft loss [14,15]. Therefore, it might be necessary to consider not only the GC sensitivity but also the original disease of the recipients. In addition, although this study only targeted Japanese people, it might also be necessary to investigate race. For example, Black recipients are known to have a considerably greater immunologic risk compared to nonblack recipients. Previous studies have shown that Black people have more HLA polymorphisms [16] and immune hyper-responsiveness [17], which make them immunologically high risk. There are various opinions on the safety and efficacy of GC reduction/withdrawal for Black people [18,19,20,21], and individual lymphocyte sensitivity to GCs would be a useful and reliable biomarker when carrying out safe and efficient immunosuppressive therapy in renal transplantation for Black people.

## 5. Conclusions

In short-term renal transplantation, individual lymphocyte sensitivity to GCs is a reliable biomarker to predict clinical outcomes after GC reduction/withdrawal. It will be possible to select patients who can have their dose of GCs decreased or withdrawn after transplantation by monitoring an individual’s lymphocyte sensitivity to GCs before transplantation.

## Figures and Tables

**Figure 1 jcm-10-01670-f001:**
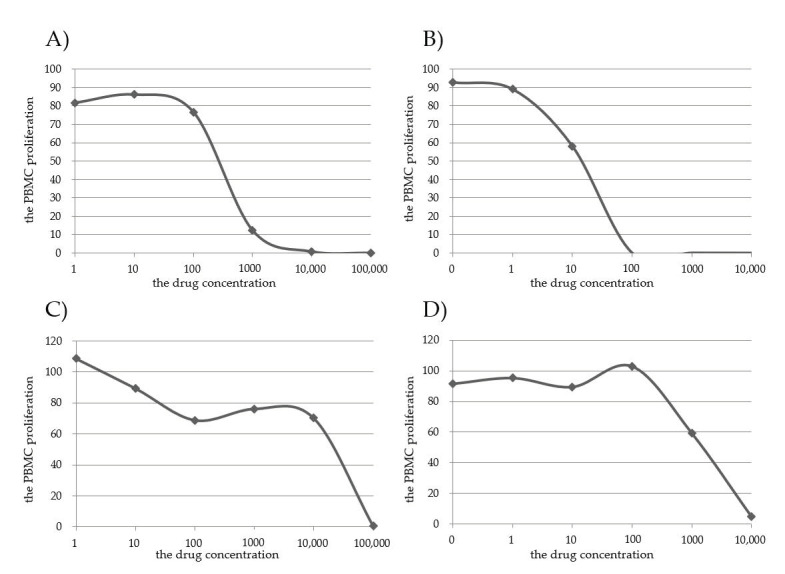
Examples of the concentration–response curve of the PBMC high and low sensitivity groups to GC. The concentration–response curve of the PBMC high-sensitivity group to COR (**A**) and to MPSL (**B**); the PBMC low-sensitivity group to COR (**C**) and to MPSL (**D**).

**Figure 2 jcm-10-01670-f002:**
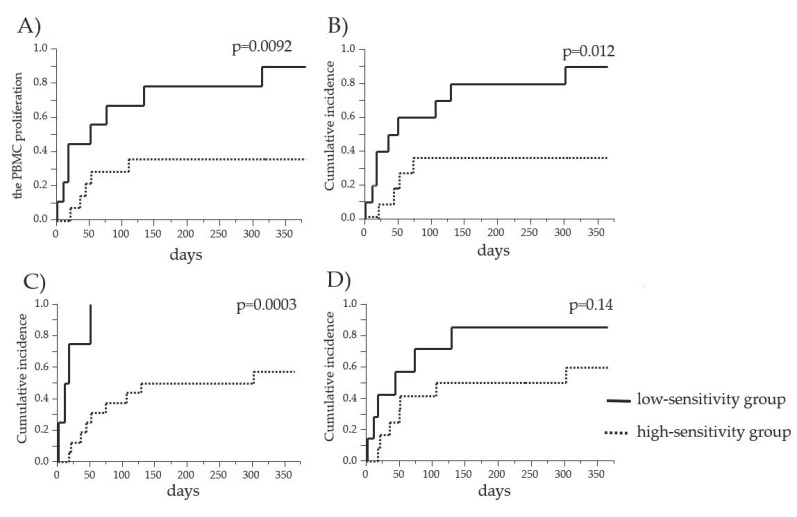
Comparison of the cumulative incidence of GC increase/resumption between the PBMC high-sensitivity group and the PBMC low-sensitivity group to GCs. The cumulative incidences of the increase/resumption of GC administrations in the PBMC high-sensitivity group and the PBMC low-sensitivity group to COR before transplantation (**A**), MPSL before transplantation (**B**), COR after transplantation (**C**), and MPSL after transplantation (**D**) were estimated. The solid lines and dashed lines indicate the data for the low-sensitivity and high-sensitivity groups, respectively.

**Figure 3 jcm-10-01670-f003:**
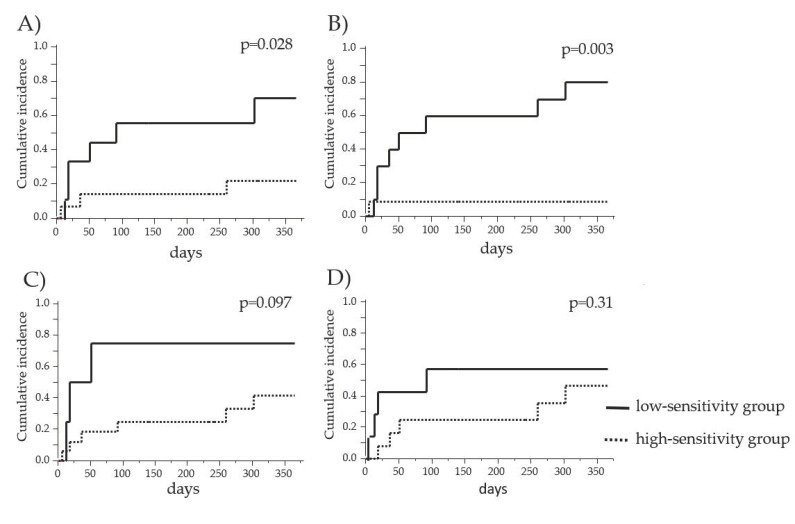
Comparison of the cumulative incidence of AR between the PBMC high-sensitivity group and the PBMC low-sensitivity group to GCs. The cumulative incidences of AR in the PBMC high-sensitivity group and the PBMC low-sensitivity group to COR before transplantation (**A**), MPSL before transplantation (**B**), COR after transplantation (**C**), and MPSL after transplantation (**D**) were estimated. The solid lines and dashed lines indicate the data for the low-sensitivity and high-sensitivity groups, respectively.

**Figure 4 jcm-10-01670-f004:**
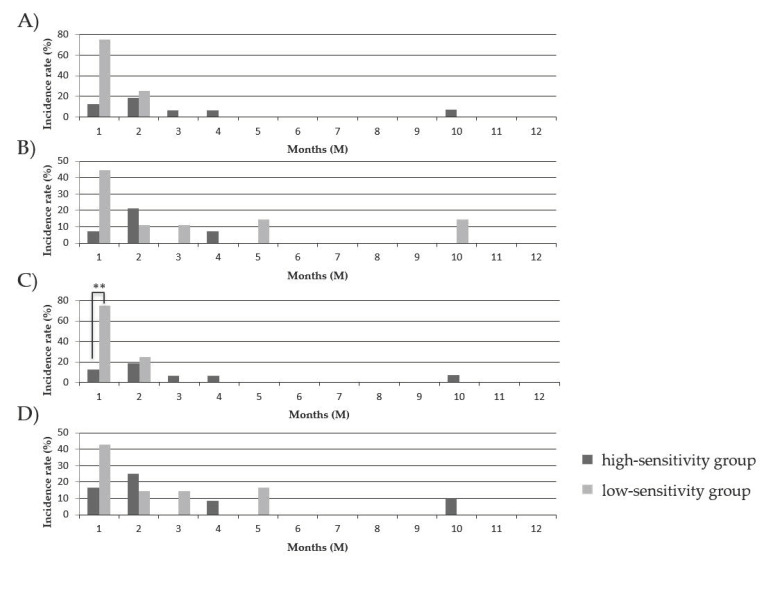
Comparison of the onset time of GC increase/resumption between the PBMC high-sensitivity group and the PBMC low-sensitivity group to GCs. The onset times of the increase/resumption of GC administrations in the PBMC high-sensitivity group and the PBMC low-sensitivity group to COR before transplantation (**A**), MPSL before transplantation (**B**), COR after transplantation (**C**), and MPSL after transplantation (**D**) were estimated. The statistical significance is presented as (**) for *p* < 0.05.

**Figure 5 jcm-10-01670-f005:**
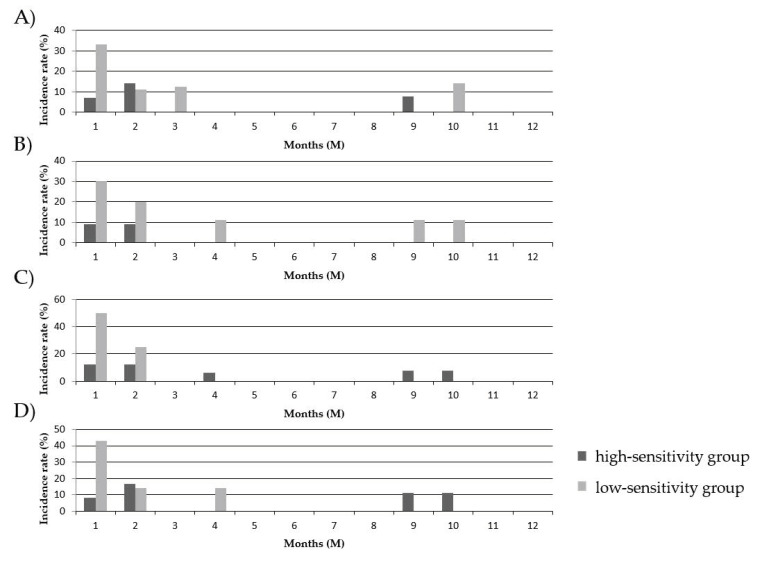
Comparison of the onset time of AR between the PBMC high-sensitivity group and the PBMC low-sensitivity group to GCs. The onset times of AR in the PBMC high-sensitivity group and the PBMC low-sensitivity group to COR before transplantation (**A**), MPSL before transplantation (**B**), COR after transplantation (**C**), and MPSL after transplantation (**D**) were estimated.

**Figure 6 jcm-10-01670-f006:**
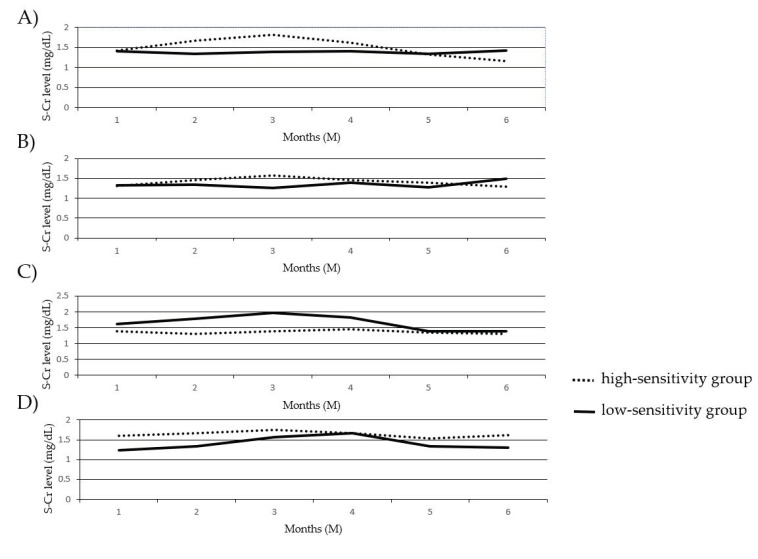
Comparison of the change in S–Cr levels between the PBMC high-sensitivity group and the PBMC low-sensitivity group to GCs. The changes in S–Cr levels in the PBMC high-sensitivity group and the PBMC low-sensitivity group to COR before transplantation (**A**), MPSL before transplantation (**B**), COR after transplantation (**C**), and MPSL after transplantation (**D**) were estimated. The solid lines and dashed lines indicate the data for the low-sensitivity and high-sensitivity groups, respectively.

**Table 1 jcm-10-01670-t001:** The recipient characteristics

	Mean ± SD or Number
Age (yr)	47.5 ± 12.0
Male/female	17/7
Body weight (kg)	62.3 ± 14.7
First/second graft	24/0
Living related/cadaveric donor	23/1
ABO blood type compatible/incompatible	22/2
The number of HLA mismatches	2.7 ± 1.1
Immunosuppressive drugs	
G-Cs (MPSL/PSL)	24/0
CNI (CYA/TAC)	14/10
Antimetabolites (MMF/MIZ/AZ + MIZ)	18/5/1
Monoclonal antibody (Bx/Bx + Rx)	22/2

HLA: Human Leukocyte Antigen; CNI: calcineurin inhibitor; MPSL: methylprednisolone; CYA: cyclosporine; TAC: tacrolimus; MMF: mycophenolate mofetil; MIZ: mizoribine; AZ: azathioprine; Bx: basiliximab; Rx: rituximab.

**Table 2 jcm-10-01670-t002:** Comparison of the 50% of PBMC proliferation (IC_50_) values between the clinical event group and the event-free group regarding experience of increase/resumption of GCs

Time Point	GCs	The Clinical Event Groupthe IC_50_ Value (IC_25_, IC_75_) ng/mL		The Event-Free Groupthe IC_50_ Value (IC_25_, IC_75_) ng/mL		*p*-Value
Before transplantation	COR	5758 (171, 11,727)	*n* = 13	93 (51, 358)	*n* = 9	0.042
	MPSL	164.5 (12.0, 540.9)	*n* = 13	2.46 (2.2, 4.9)	*n* = 7	0.0049
After transplantation	COR	620(55, 6349)	*n* = 13	62 (35, 524)	*n* = 6	0.19
	MPSL	19.8 (5.9, 44.4)	*n* = 13	4.0 (0.1, 49.5)	*n* = 5	0.18

The median IC_50_ values against mitogen-activated proliferation of PBMCs were shown. In parentheses, the median of IC_25_ and IC_75_ values are shown from left to right, respectively.

**Table 3 jcm-10-01670-t003:** Comparison of the IC_50_ values between the clinical event group and the event-free group regarding the incidence of acute graft rejection (AR)

Time Point	GCs	The Clinical Event Group the IC_50_ Value (IC_25_, IC_75_) ng/mL		The Event-Free Group the IC_50_ Value (IC_25_, IC_75_) ng/mL		*p*-Value
Before transplantation	COR	9529 (255, 12,884)	*n* = 9	93 (51, 495)	*n* = 11	0.011
	MPSL	175.6 (44.4, 950.9)	*n* = 9	3.0 (2.3, 153.8)	*n* = 9	0.0071
After transplantation	COR	620 (151, 8751)	*n* = 9	62 (26, 820)	*n* = 8	0.083
	MPSL	19.8 (14.1, 305.3)	*n* = 9	4.0 (0.1, 29.1)	*n* = 7	0.08

The median IC_50_ values against mitogen-activated proliferation of PBMCs are shown. In parentheses, the median IC_25_ and IC_75_ values are shown from left to right, respectively.

## Data Availability

The data presented in this study are available on request from the corresponding author.

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
