# Peer review of "Individual Lymphocyte Sensitivity to Steroids as a Reliable Biomarker for Clinical Outcome after Steroid Withdrawal in Japanese Renal Transplantation"

_jcm, 2021, doi:10.3390/jcm10081670_

Round 1

Reviewer 1 Report

Individual lymphocyte sensitivity to steroids as a reliable
biomarker for clinical outcome after steroid withdrawal in
renal transplantation.

1. Most of the patients included in this study were on
multiple immunosuppressive drugs in addition to
Methylprednisolone (MPSL). Thus, peripheral blood
mononuclear cells taken from these patients for the
study were also exposed to immunosuppressive drugs
other than MPSL. It is not clear whether the
immunosuppressive drugs other than MPSL have
somehow contributed to the categorization of patients
in high and low sensitivity groups. The data should be
analysed and outcome reported accordingly.
2. Reference numbering may be checked – first two
references have been given number 1.
4. It may be considered to add ‘ Japanese’ in the title to
emphasize the ethnicity of the patients.

Author Response

Journal of Clinical Medicine

April 9, 2021

Dear Editor

Thank you very much for your messages of April 6, 2021, regarding our manuscript entitled “Individual lymphocyte sensitivity to steroids as a reliable biomarker for clinical outcome after steroid withdrawal in Japanese renal transplantation. (Authors: Masaaki Okihara, Hironori Takeuchi, Yukiko Kikuchi, Isao Akashi, Yu Kihara, Osamu Konno, Hitoshi Iwamoto, Takashi Oda, Sachiko Tanaka, Sakae Unezaki, Toshihiko Hirano, Manuscript ID: jcm-1157360)” along with the comments of the reviewers.

We revised the manuscript according to the comments of the reviewers. The revised points were colored (red characters) in the revised manuscript. Please see the uploaded files to accommodate the reviewer’s comments. Our manuscript has been read and corrected by the colleague who is proficient in English before this submission.

Trusting favorable consideration for the acceptance of our revised manuscript.

Sincerely yours,

Toshihiko Hirano

Department of Clinical Pharmacology

Tokyo University of Pharmacy and Life Sciences

1432-1 Horinouchi, Hachioji

Tokyo 192-0392, Japan.

Tel: +81-42(676)5794, Fax: +81-42(676)5798

E-mail: hiranot@toyaku.ac.jp

1. Most of the patients included in this study were on multiple immunosuppressive drugs in addition to Methylprednisolone (MPSL). Thus, peripheral blood mononuclear cells taken from these patients for the study were also exposed to immunosuppressive drugs other than MPSL. It is not clear whether the immunosuppressive drugs other than MPSL have somehow contributed to the categorization of patients in high and low sensitivity groups. The data should be analyzed and outcome reported accordingly.

The immunosuppressive drugs other than MPSL might affect the peripheral blood mononuclear cells and the sensitivity to GC, as the reviewer pointed out. However, the difference of immunosuppressive drugs other than MPSL between the PBMC high-sensitivity group and the low-sensitivity group to COR and MPSL before and after transplantation was not statistically significant in all cases. Thus, although the immunosuppressive drugs other than MPSL might affect the sensitivity to GC, we believe that these results were credible. In addition, no significant difference was observed between two recipient groups in other recipient characteristics, and we added to Results, as shown in 3.1. the recipient characteristics. (p.5, lines 70-74 of the revised manuscript)

2. Reference numbering may be checked – first two references have been given number 1.

4. It may be considered to add ‘ Japanese’ in the title to emphasize the ethnicity of the patients.

We revised the manuscript according to the comments and suggestions of the reviewer.

Reviewer 2 Report

The sensitivity of transplantation patients to glucocorticoid treatment was determined before and after transplantation by measuring the IC50 of methylprednisolone and cortisol in Con A induced PBMCs. The incidence of acute rejection in the first year after transplantation was higher in patients with low glucocorticoid sensitivity compared to patients with high sensitivity.

The authors clearly describe the aim and conclusion of the study, but the English language, introduction and methods of the manuscript can be improved.

Introduction

  • Please check the use of language, especially section 3 (line 54-62)
  • Reference missing at line 54 that describes lack of PK/PD relationship in glucocorticoid treatment
  • The abbreviation S-Cr is not spelled out

Methods 

  • Include information on how many hours/days before and after transplantations the PBMC samples were taken.
  • How was the cutoff value for the groups 'high sensitive' and 'low sensitive' determined? Reference [12] does not seem appropriate here
  • How was the IC50 calculated? What model was used to interpolate the dose-response data? Representative plots could be added as supplemental data. 

Results

  • It is advised to include the dose-response curves (proliferation rate vs CG concentration) of the different subgroups (event, event-free, high-sensitivity and low-sensitivity), for better visualization of the differences in IC50.

Author Response

Journal of Clinical Medicine

April 9, 2021

Dear Editor

Thank you very much for your messages of April 6, 2021, regarding our manuscript entitled “Individual lymphocyte sensitivity to steroids as a reliable biomarker for clinical outcome after steroid withdrawal in Japanese renal transplantation. (Authors: Masaaki Okihara, Hironori Takeuchi, Yukiko Kikuchi, Isao Akashi, Yu Kihara, Osamu Konno, Hitoshi Iwamoto, Takashi Oda, Sachiko Tanaka, Sakae Unezaki, Toshihiko Hirano, Manuscript ID: jcm-1157360)” along with the comments of the reviewers.

We revised the manuscript according to the comments of the reviewers. The revised points were colored (red characters) in the revised manuscript. Please see the uploaded files to accommodate the reviewer’s comments. Our manuscript has been read and corrected by the colleague who is proficient in English before this submission.

Trusting favorable consideration for the acceptance of our revised manuscript.

Sincerely yours,

Toshihiko Hirano

Department of Clinical Pharmacology

Tokyo University of Pharmacy and Life Sciences

1432-1 Horinouchi, Hachioji

Tokyo 192-0392, Japan.

Tel: +81-42(676)5794, Fax: +81-42(676)5798

E-mail: hiranot@toyaku.ac.jp

  1. Introduction

It has been well known that there could be no high relationship between blood concentration and therapeutic efficacy of GCs. Although we described in original manuscript that there could be no relationship between blood concentration and therapeutic efficacy of GCs, actually, it was correct that it was not unrelated, but poorly related. Thus, we revised the manuscript. (p. 2, lines 54 of the revised manuscript)

  1. Methods

We added to p.3, lines 98-99 of the revised manuscript due to lack of information on how many hours/days before and after transplantations the PBMC samples were taken.

We revised the reference numbers because these were incorrect, which made it difficult to understand. The cutoff values were calculated based on our previous study [12] and we defined that the cut-off values of COR and MPSL were 3579.98 ng/mL and 21.5 ng/mL, respectively, as described in the manuscript.

We calculated the IC50 from the concentration-response curve and interpolated the data using the Emax model. (p.3, lines 122-125 of the revised manuscript) In addition, according to the suggestion of the reviewer, we newly added the representative plots as supplemental data, as shown in Fig. 1. (p.4, lines 134-140 of the revised manuscript)

  1. Results

It will be better for us, as the reviewer suggested, to show the curves for better visualization of the differences in IC50. However, we considered that the IC50 S were an established evaluation method and the variation of IC50S could sufficiently show in the interquartile range. Therefore, we did not show the dose-response curves of the different subgroups. In addition, abandoned due to limited time to submission. We are very sorry that we could not make effective use despite the meaningful suggestion of the reviewer.
